

# Feeding ecology of invasive lionfish in the Punta Frances MPA, Cuba: insight into morphological features, diet and management

Laura del Río[1,*], Zenaida María Navarro-Martínez[1,*], Alexei Ruiz-Abierno[1], Pedro Pablo Chevalier-Monteagudo[2], Jorge A. Angulo-Valdes[3] and Leandro Rodriguez-Viera[1]

[1] Center for Marine Research, University of Havana, La Habana, Cuba
[2] National Aquarium of Cuba, La Habana, Cuba
[3] Eckerd College, St. Petersburg, Florida, United States
[*] These authors contributed equally to this work.

Corresponding author
Leandro Rodriguez-Viera,
leokarma@gmail.com

## ABSTRACT

Cuba's shelf has been invaded by lionfish (*Pterois volitans/Pterois miles*), which have become established over the archipelago, including areas of natural importance. The present study aims to evaluate morphometric features of lionfish and to explore the relationship between lionfish size and diet composition in different habitats in the Punta Frances National Park, Cuba. In total 620 lionfish were captured at 29 sites between 2013 and 2016. Lionfish stomachs were removed and their contents were analyzed using frequency and numerical methods. The length-weight allometric relationship was obtained, and a decrease in lionfish sizes was shown over time, likely due to the extractions carried out. The diet was composed by fishes, crustaceans, mollusks and phytobenthos, with a predominance of fishes. Lionfish caught in seagrass beds tended to be smaller in size and consumed fewer fishes and more crustaceans than those captured in coral reefs. A positive correlation was observed between lionfish body size and gape size; however, no significant correlation was detected between lionfish body size and prey size. Larger lionfish tended to consume more fishes, while crustaceans were more significant in the diet of juvenile lionfish. This is the first study that examines the feeding habits of lionfish in the Punta Frances MPA, and provides valuable information on lionfish inhabiting this MPA across four years of sampling. Furthermore, this research may serve as a baseline for subsequent evaluations of lionfish impact and management actions in the area.

## INTRODUCTION

Invasive marine species are a major global problem, which requires regional and global action to attain suitable solutions (*Bax et al., 2003*). Lionfish are the first non-native marine fishes that successfully invaded the western Atlantic Ocean and the Caribbean Sea

(*Schofield, 2010*). Two species are reported: *Pterois volitans* (Linnaeus 1758) and *Pterois miles* (Bennett 1828). Both of them were accidentally, or intentionally, introduced outside their natural range, and they have been able to successfully maintain populations in the western Atlantic and the Caribbean Sea (*Schofield, 2009*, *2010*). The lionfish introduction and dispersal have the potential to threaten biological diversity, causing damage to the environment, the economy, and human health (*Kizer, McKinney & Auerbach, 1985*; *Morris & Akins, 2009*; *Albins, 2015*; *Cobián-Rojas et al., 2018*). Thereby, they present the typical characteristics that make them invasive alien species (*Bax et al., 2003*; *Mendoza-Alfaro et al., 2011*).

The first report on lionfish in the western Atlantic region was made in Florida (USA) in 1985 (*Morris & Akins, 2009*); other reports followed in the Bermuda Islands and Bahamas (*Whitfield et al., 2002*; *Schofield, 2009*; *Morris & Akins, 2009*). Lionfish presence in Cuban waters was first recorded on the north coast in 2007, and sightings and captures occurred in several locations throughout the country during a year (*Chevalier et al., 2008*). Lionfish presence was also documented in other areas of Latin America and the Caribbean in the following years (*e.g.*, Dominican Republic (*Guerrero & Franco, 2008*); Colombia (*González et al., 2009*; *Bustos-Montes et al., 2020*); Cayman Islands, Jamaica, Puerto Rico, Haiti, Belize, Panama, Honduras, Costa Rica (*Schofield, 2009*); Nicaragua (*Schofield, 2010*); Gulf of Mexico (*Aguilar-Perera & Tuz-Sulub, 2010*); and Brazil (*Ferreira et al., 2015*)).

In Cuba, the presence of lionfish has been reported inside marine protected areas (MPAs) such as the Punta Frances National Park (PFNP), in the southwest region of the Isla de la Juventud. This MPA is recognized by its coral reef, which is one of its predominant habitats, and has been used for tourism purposes (mostly SCUBA diving) since 1976 (*Angulo-Valdes et al., 2007*). In addition, this MPA contains mangroves and seagrass beds, which are typical Caribbean coastal habitats that provide great natural value to the area (*de la Guardia et al., 2004*; *de la Guardia, González-Díaz & Iglesias, 2004*; *Navarro-Martínez, 2015*; *Rodríguez-Viera et al., 2017*). The presence of this interconnected system ensures a diversity of biotopes that is conducive to the establishment of highly diverse and complex marine communities. Despite high connectivity in PFNP, each of these three habitats exhibit unique characteristics that could modulate lionfish abundance and survival.

Most studies on lionfish have been focused on their diet, because their principal impact on the invaded area is through direct predation on fishes and marine invertebrates, or competition for food with organisms of similar trophic level. Lionfish diet composition study allows knowing which are the most sensitive organisms affected by predation or competition; and concentrating efforts on the evaluation of lionfish impact on marine communities. The morphometric variables such as total length is an important parameter to take into account, and it has been the subject of several studies in the region (*Frazer et al., 2012*; *Claydon, Calosso & Traiger, 2012*; *de la Guardia et al., 2017*; *Barbour et al., 2011*; *Darling et al., 2011*). For instance, the comparison of lionfish length between different ecosystems provide information about habitat changes during their ontogenetic development (*Claydon, Calosso & Traiger, 2012*; *Pimiento et al., 2015*; *de la Guardia et al., 2017*). On the other hand, the analysis of the total length temporal variation in sites where

 

lionfish extractions are carried out can show if these extractions are efficient in their control (*Akins, 2013*; *Frazer et al., 2012*). In addition, the analysis of the relationship between lionfish total length and diet allows evaluating the changes in the type of prey consumed by juveniles and adults (*Morris & Akins, 2009*; *McCleery, 2011*; *Muñoz, Currin & Whitfield, 2011*). Using the recorded total length and species-specific length to weight allometric relationship for lionfish in the region, the weight and subsequently the extracted biomass can be estimated (*Darling et al., 2011*). All these findings can be greatly helpful when developing control strategies; and despite the well-studied condition of lionfish, there are still important gaps in the ecology and management of lionfish in the invaded region.

Lionfish presence could threaten the ecological system stability that exists in PFNP, by means of direct predation and competition with native fish species (*Albins & Hixon, 2008*; *Morris & Akins, 2009*; *Cobián-Rojas et al., 2018*). This can provoke a decrease in the attractiveness of the reefs and consequently affect the development of tourist activities that take place in the area and depends considerably on the attractiveness of the fauna and the ecosystem as a whole, *e.g.*, recreational diving. The knowledge about the lionfish effect on the reef trophic webs, as well as adequate control of this invader, are essential for the ecosystem conservation. Despite the potential risk of losing economic revenues from tourism activities due to the lionfish presence, no study has been done to address its possible impacts on the mangrove-seagrass bed-coral reef system of PFNP. Therefore, this paper aims to evaluate morphometric features (size distributions per habitat, weight-length relationships) of lionfish, and to explore the relationship between lionfish size and diet composition in different habitats in PFNP, Cuba. Information acquired during four years of sampling also constitutes a baseline for future studies addressing lionfish impacts in the area. The knowledge generated contributes to management recommendations for these species in PFNP, its adjacent areas, and other similar MPAs.

## MATERIALS AND METHODS

### Study area

Punta Frances National Park is located in the southwest portion of Isla de la Juventud, Cuba, specifically on the Carapachibey Peninsula. It encompasses an area of 4,598 ha, of which 3,036 ha are marine areas (*Perera-Valderrama et al., 2018*). Samplings took place within PFNP, where coral reefs are one of the dominant habitats, and also in adjacent areas with predominance of seagrass bed (Fig. 1). The coral reefs exhibit a high spatial heterogeneity characterized by well-developed back and fore reefs including a large spur and groove area, and a notable drop off (*de la Guardia et al., 2004*). Lionfish were captured at 29 sites, which are located within and adjacent to PFNP. Sampling depth was variable depending on habitat and site; it range mostly varied between 5–23 m and the maximum depth was ca. 30 m. For analyses purposes, the sites were grouped into three main sampling regions (R1, R2 and R3) according to the habitats and their influence on each region (*i.e.*, mangroves, seagrass bed, coral reefs) (Fig. 1, Table 1). This artificial grouping allows a better understanding of habitat influence over lionfish sizes and diet. The R1 and R2 regions are located in coral reef areas within PFNP, but R2 shows higher

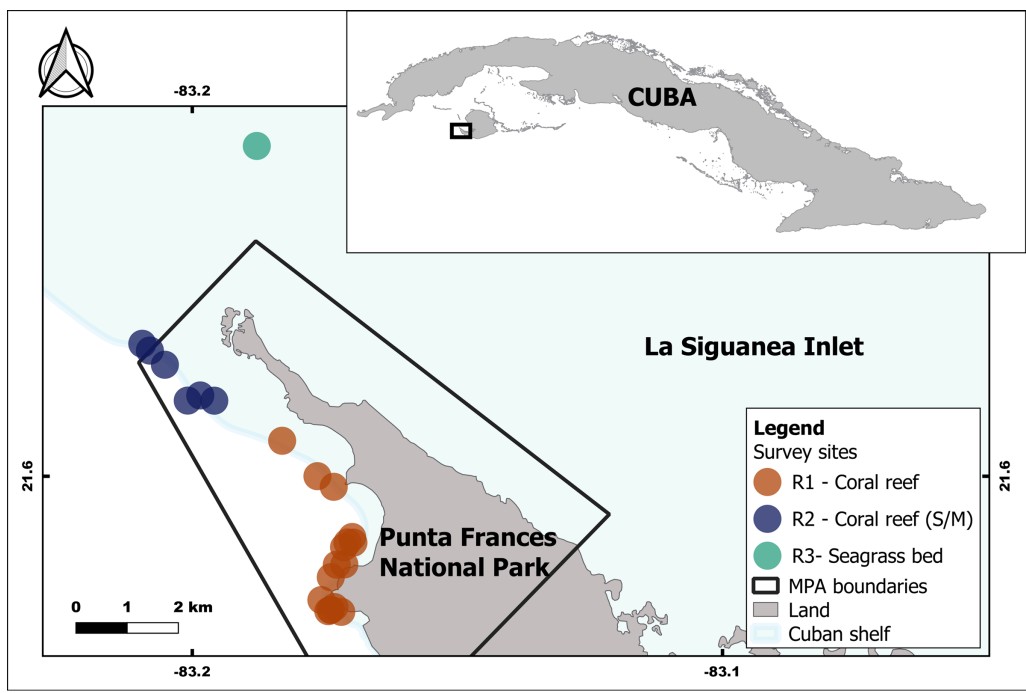

**Figure 1 Geographic location of the study area inside and associated with the Punta Frances National Park (delimited by the polygon) and the sampling sites grouped into three regions: R1, R2 and R3.** This artificial grouping allows a better understanding of habitat influence over lionfish sizes and diet. The R1 and R2 regions are located in coral reef areas, but R2 shows higher influence of mangrove and seagrass areas. The R3 region is seagrass bed habitat.

**Table 1 Number of lionfish caught by each region in the Punta Frances National Park, during the 4 years of sampling (2013–2016).**

| Region | Influence of habitats | | | Lionfish caught | | | | |
|---|---|---|---|---|---|---|---|---|
| | Coral reef | Seagrass | Mangroves | 2013 | 2014 | 2015 | 2016 | Total |
| R1 | M | | | 16 | 70 | 226 | 95 | 407 |
| R2 | M | A | A | 44 | 24 | 16 | 37 | 121 |
| R3 | | M | | NS | NS | 62 | NS | 62 |
| UR | | | | | | 9 | 21 | 30 |
| Total | | | | 60 | 94 | 313 | 153 | 620 |

**Note:**
NS, Not sampled; M, Main habitat; A, Associated habitat of great influence; UR, Unidentified Region.

influence of mangrove and seagrass areas. The R3 region is seagrass bed habitat located outside PFNP, and it was only sampled in 2015. PFNP fourth group "Unidentified Region" (UR) was created to account for all individual lionfish and stomach content samples that lost the labels due to improper storage and management.

## Sampling methodology

Lionfish were collected by SCUBA diving, using pole spears and following the protocol from *Chevalier et al. (2014)* for the study of lionfish in Cuba. Lionfish were collected in

**Table 2 Methods used to analyze the stomach content of lionfish in Punta Frances.**

| Method | Equation | References |
|---|---|---|
| Numerical method (%N) | $\%N = 100 * N_i/N$ | *Hyslop (1980)* |
| | $N_i$: number of entities of each food category present in a stomach or region | |
| | N: total number of entities in a stomach or region | |
| Frequency method or frequency of occurrence (%F) | $\%F = 100 * f_i/TS$ | *Rosecchi & Novaze (1987)* |
| | $f_i$: number of stomachs where the entity was found | |
| | TS: total number of stomachs analyzed for each region | |

Punta Frances MPA, Cuba, under all applicable local, state, and Cuban laws regulations; and the regulation of the Committee of the Center for Marine Research at the University of Havana for animal care and use, Authorization code CIM/029. Captures were made during June, July and August each year from 2013 to 2016 (Table 1).

Total length of freshly caught lionfish was recorded using an ichthyometer (accurate to 0.1 mm). Additionally, for lionfish collected in 2015, gape size was estimated by recording the width and height of the mouth, to calculate gape area; and lionfish weight was recorded using a portable digital scale (Ohaus CS200; error ± 0.1 g). For all lionfish samples, the stomachs were removed and contents were visually analyzed on site or preserved in 90% ethanol and analyzed later in the laboratory. A dissecting microscope was used for diet analysis, when necessary. Prey items found inside each stomach were classified to the lowest possible taxonomic level, according to *Guitart (1985)*, *Carpenter (2002)* and *Humann & Deloach (2006)* for fishes; *Abele & Kim (1986)* for crustaceans; and *Abbott (1974)* for mollusks. Additionally, prey items were grouped into four categories: fishes, crustaceans, mollusks and phytobenthos. Prey size from lionfish collected in 2015 was measured as standard length for fishes, cephalothorax width and length for crustaceans using a ruler, and conch length for mollusks using a vernier caliper following the criteria of *Abbott (1974)*.

## Stomach content analysis

The frequency of occurrence of the taxa identified in the lionfish stomachs was calculated as the number of times a certain prey item appears within the total number of stomachs analyzed. Additionally, stomach contents of individual lionfish were analyzed, using the frequency method (%F) and the numerical method (%N) (Table 2; *Hyslop, 1980*; *Rosecchi & Novaze, 1987*). The frequency method takes into account the number of stomachs in which a given item is found in relation to the total number of stomachs analyzed in a given area, in this case it was analyzed at the regional level (*i.e.*, R1, R2 and R3). The numerical method allows knowing the representativeness of a certain item in a stomach, by calculating the percentage that represents the number of individuals of that food category with respect to the total number of individuals found in that stomach. Lionfish with empty stomach were excluded from the analyses.

## Statistical analyses

Temporal (2013–2016) and spatial (R1, R2, R3) variation of the lionfish total length was compared using mean values ±95% confidence interval. In addition, for testing differences in lionfish mean length per year between regions, a Mann-Whitney U test was carried out except for the year 2015, for which a Kruskal-Wallis ANOVA by Ranks test was carried out because three regions were sampled that year. A Kruskal-Wallis ANOVA by Ranks test was also made for lionfish length comparisons per region between years. All significant effects were examined further with multiple comparisons if more than two sample groups. For all tests, significance was indicated when $p \leq 0.05$.

The length-weight relationship for lionfish was calculated using data from lionfish captured during June, July, and August, 2015. With these data, a scatter plot was constructed and adjusted according to the equation:

$$W = a * L^b$$

where $W$ represents weight (g), $L$ represents total length (mm), and $a$ and $b$ are species-specific constants (*Froese, 2006*). The former equation provided the constants $a$ and $b$ for lionfish in the area, and thereby allows to estimate the weight of the specimens caught in 2013, 2014 and 2016 (and therefore the extracted biomass), considering these former lionfish were measured but not weighted.

The consumption of fishes and crustaceans was compared per sampling region using the frequency and numerical method. For this analysis, we excluded smaller and larger lionfish with sizes not present in all the three regions for avoiding a combining effect of both habitat and lionfish size (*i.e.*, effect of habitat over size and then size over diet *vs* direct effect of habitat over diet). Therefore, only lionfish within the size range sampled in the three regions (100–352 mm, total length) were included, which allow to analyze only the effect of habitat type and avoid that differences in size between regions could influence the result. The relationships between the lionfish's mouth dimensions and its total length, and between lionfish total length and the average sizes of its prey, were analyzed through Spearman rank correlations.

Due to logistic reasons some lionfish were not fully processed, thereby the sample size was not equal in all analyses. The software package Statistica 7 (*StatSoft, Inc, 2004*) was used for all statistical tests performed and graphs construction.

## RESULTS

### Morphometric features

#### Total length: temporal and spatial variation

Overall 620 lionfish were captured in this study (Table 1). Mean lionfish total length obtained from all regions was 250.47 mm (CI [245.05–255.88] mm, $n = 588$ lionfish), with a maximum length of 405 mm and a minimum length of 87 mm (Fig. 2; Table 3). In R3, a greater number of juvenile lionfish was observed, while in R1 and R2 predominated lionfish with sizes greater than sexual maturity (160 mm) (*Froese & Pauly, 2019*) (Fig. 2). Mean lionfish size decreased over time which was most evident in R1 (Fig. 3; Table 3).

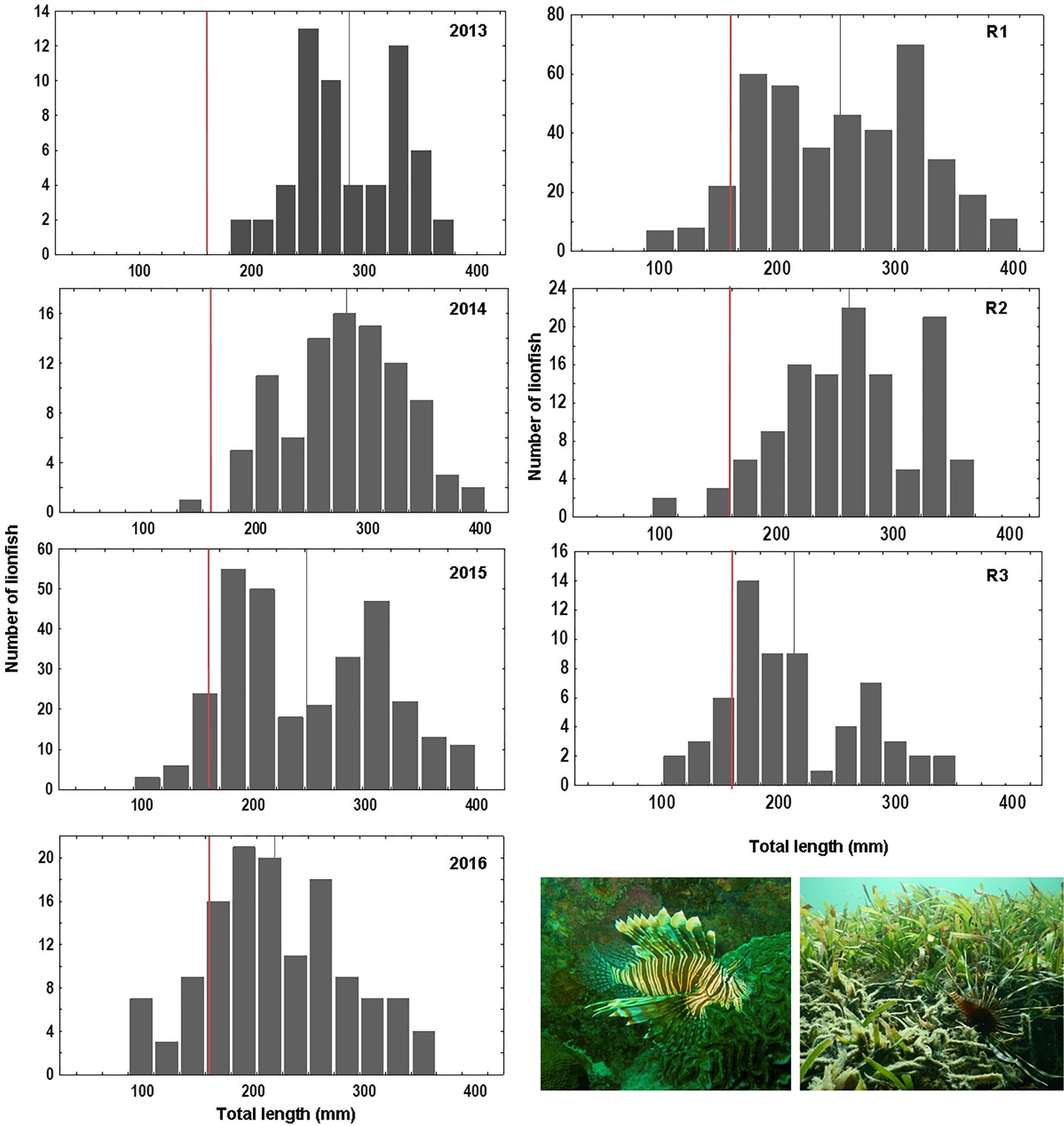

**Figure 2 Length frequency of lionfish (total length, mm) caught by year: 2013 ($n$ = 59), 2014 ($n$ = 94), 2015 ($n$ = 303), 2016 ($n$ = 132) and by sampling region: R1 ($n$ = 406), R2 ($n$ = 120), R3 ($n$ = 62).** Vertical grey line indicates mean value of lionfish size per year/region; vertical red line indicates length at mature: ~160 mm, based on *Froese & Pauly (2019)*.

**Table 3 Mean total length and total biomass of lionfish caught by each region in the Punta Frances National Park (R1, R2, R3), during the four years of sampling (2013–2016).**

| Region | Years | | | | |
|---|---|---|---|---|---|
| | **2013** | **2014** | **2015** | **2016** | **Total** |
| | **Mean total length (mm) [95% confidence interval, mm], *n* = 588** | | | | |
| R1 | 299.5 [277.02–321.98] | 278.39 [265.21–291.56] | 259.52 [250.52–268.51] | 210.63 [198.03–223.23] | 252.91 [246.21–259.60] |
| R2 | 281.77 [267.05–296.48] | 286.96 [267.33–306.58] | 217.50 [187.65–247.35] | 241.30 [220.79–261.81] | 261.76 [251.17–272.35] |
| R3 | NS | NS | 212.63 [197.99–227.28] | NS | 212.63 [197.98–227.28] |
| Total | 286.58 [274.41–298.74] | 280.57 [269.71–291.44] | 247.70 [239.94–255.46] | 219.23 [208.37–230.08] | 250.47 [245.05–255.88] |
| | **Total biomass (g), *n* = 620**\*\* | | | | |
| R1 | 5.86 | 21.64 | 67.73\* | 14.07 | 109.30 |
| R2 | 13.59 | 7.83 | 2.59\* | 7.85 | 31.86 |
| R3 | NS | NS | 8.72\* | NS | 8.72 |
| UR | NS | NS | 2.08\* | 2.37 | 4.45 |
| Total | 19.45 | 29.47 | 81.12 | 24.29 | 154.33 |

**Notes:**
NS, Not sampled; UR, Unidentified Region, non-included on mean total length comparisons.
\* Biomass values obtained from the weight recorded *in situ* with the portable balance (except by one value), the rest of the values were calculated from the weight estimated by the weight–length equation (W = 6 * $10^{-6}$ $L^{3.15}$) in *P. volitans/P. miles* obtained in the present study.
\*\* Including lionfish caught in UR (Unidentified Region).
Heading lines appear in bold.

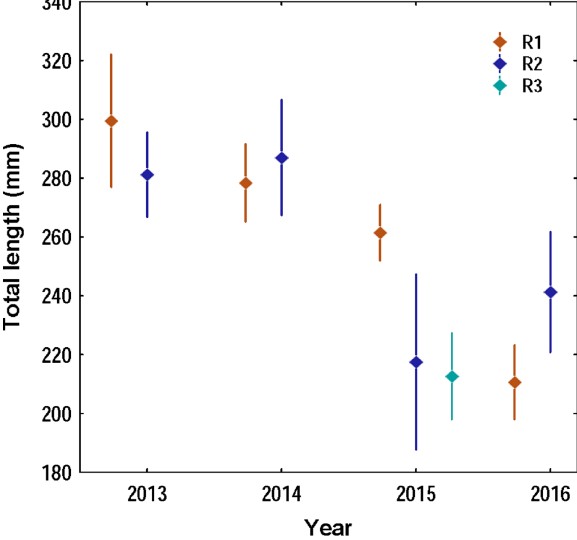

**Figure 3 Variation of the total length of the lionfish in the three capture regions during the four years of study in the Punta Frances National Park.** The points represent the means of the total lengths and the vertical lines the confidence interval (95%).

In 2013 and 2014, lionfish sizes were not significantly different between R1 and R2 regions (mean sizes from 2013 (R1: 299.50 mm and R2: 281.77 mm) and 2014 (R1: 278.39 mm and R2: 286.96 mm)). No differences were found neither in lionfish size from 2015 between R2 (mean size: 217.50 mm) and the other regions (mean size (R1: 259.52 mm and R3: 212.63 mm)), but lionfish sizes between R1 and R3 were significantly different. In addition,

**Table 4 Statistical information obtained from the non-parametric tests: Mann-Whitney U test (M-W) and Kruskal-Wallis ANOVA test (K-W) and the multiple comparisons test carried out for lionfish total length comparisons.**

| Non-parametric tests | | | | Multiple comparisons test (p-values) | | | | | | | |
|---|---|---|---|---|---|---|---|---|---|---|---|
| **Years** | **Test** | **Parameter** | **p-value** | **R1** | **2014** | **2015** | **2016** | **R2** | **2014** | **2015** | **2016** |
| 2013 | M-W | U = 280.00 | 0.27 | **2013** | 1 | 0.12 | <0.01 | **2013** | 1 | <0.01 | 0.03 |
| 2014 | M-W | U = 773.00 | 0.56 | **2014** | | 0.16 | <0.01 | **2014** | | <0.01 | 0.02 |
| 2015 | K-W | H = 25.01 | <0.01 | **2015** | | | <0.01 | **2015** | | | 1 |
| 2016 | M-W | U = 1,256.50 | 0.01 | **2016** | | | | **2016** | | | |
| **Region** | **Test** | **Parameter** | **p-value** | **2015** | **R2** | **R3** | | | | | |
| R1 | K-W | H = 53.53 | <0.01 | **R1** | 0.11 | <0.01 | | | | | |
| R2 | K-W | H = 20.77 | <0.01 | **R2** | | 1 | | | | | |

**Note:**
Heading lines appear in bold. Elements for comparisons in multiple comparisons also appear in bold.

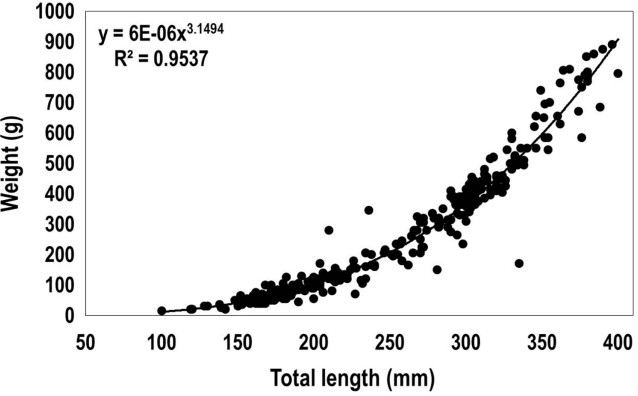

**Figure 4 Length-weight relationship obtained from 310 lionfish captured in the Punta Frances National Park (June–August, 2015).**

significant differences in lionfish size were observed between regions in 2016 (mean sizes (R1: 210.63 mm and R2: 241.30 mm)) (Fig. 3; Tables 3, 4). Within regions, lionfish size in R1 was significantly smaller in 2016 compared to the previous 3 years, and lionfish size in R2 was significantly smaller in 2015 and 2016 compared with 2013 and 2014 (Fig. 3, Table 4). The smallest mean sizes of lionfish were observed in 2015 and 2016 (Fig. 3, Table 3).

### Length-weight relationship

The relationship between the total length and weight of 310 lionfish captured in 2015 among the three sampling regions (total length ranged from 100 to 400 mm, including immature, females and males) was fit to the allometric relationship for fishes (*Froese, 2006*):

$$W = a * L^b$$

where $a = 6 * 10^{-6}$ and $b = 3.15$, $W$ (weight, g), $L$ (total length, mm) (Fig. 4). Based on the recorded biomass (from data collected in 2015) and the estimated biomass after using the obtained length-weight relationship, the total biomass of lionfish caught in this study was estimated as 154.33 kg (Table 3). The largest biomass of lionfish was extracted in R1 (109.30 kg).

## Lionfish diet characterization

### General composition

In this study, 523 stomachs were analyzed, out of these 153 were empty, representing 29.25% of the total (Table S1). A total of 1,247 prey items were identified in the stomachs. The best-represented group in the diet was that of fishes (86.69%), followed by crustaceans (12.03%), mollusks (0.40%), and phytobenthos (0.88%). A total of 24 genera and 23 species were identified among the four categories (Table 5). Among the crustaceans, species of the orders Decapoda and Stomatopoda were found. Within the Decapoda order, the Caridea infraorder was the best represented group. The most abundant orders of fishes in the diet were Labriformes and Gobiiformes.

Fourteen fish families were present in the lionfish diet. The best represented family was Labridae, followed by Grammatidae, Pomacentridae, and Scaridae (Table 5). Other families such as Serranidae, Apogonidae and Acanthuridae were registered in more than two occasions, while the rest of the fish families were represented by one or two specimens. The most abundant fish species in the lionfish stomach content were *Halichoeres bivittatus*, *Gramma loreto* and *Thalasoma bifasciatum* (Table 5).

### Diet composition by region

Both, the numerical and frequency methods showed that diet composition was dominated by fishes over crustaceans in all sampling regions, but the frequency method showed proportionally higher crustacean consumption than the numerical (Figs. 5A and 5B). Taking both methods into account, it was observed that the amount of fishes and crustaceans consumed was similar between R1 and R2. However, in R3 the consumption of fishes was proportionally lower, and that of crustaceans higher, than those found in R1 and R2.

### Relationship between diet and morphometric features

A strong positive correlation ($r = 0.94$) was found between the total length and the gape size of lionfish (Fig. 6A). However, no significant correlation ($r = 0.18$) was observed between lionfish total length and the ingested prey size (prey size range: 5–129 mm) (Fig. 6B). The predation of fishes, besides being higher than that all other categories (*i.e.*, crustaceans, mollusks and phytobenthos), increased with larger lionfish body size (Fig. 7). In contrast, crustacean ingestion was higher for smaller lionfish, and decreased with increasing total length values. No trend was identified regarding the consumption of mollusks and phytobenthos with increasing sizes. Phytobenthos was present in five stomachs, being the sole prey type in one of them (Fig. 7).

**Table 5 Taxonomic classification according to _Nelson, Grande & Wilson (2016)_ and _WoRMS Editorial Board (2019)_, and frequency of appearance (Fi) of the entities identified up to that taxonomic category within the stomachs of lionfish analyzed.**

| Taxa | Species | Fi |
|---|---|---|
| **PLANTAE KINGDOM** | | |
| **PHYLUM** Tracheophyta | | |
| **CLASS** Magnoliopsida | | |
| **Order** Alismatales | | |
| Family Hydrocharitaceae | _Thalassia testudinum_ (K.D. Koenig, 1805) | 2 |
| **PHYLUM** Rhodophyta | | |
| **CLASS** Florideophyceae | | |
| **Order** Gigartinales | | 1* |
| **ANIMALIA KINGDOM** | | |
| **PHYLUM** Arthropoda | | |
| **CLASS Malacostraca** | | |
| **Order Decapoda** | | |
| Suborder Dendrobranchiata | | 4* |
| Suborder Pleocyemata | | |
| Infraorder Caridea | | 11* |
| Family Palaemonidae | _Brachycarpus biunguiculatus_ (Lucas, 1846) | 1 |
| Family Rhynchocinetidae | _Cinetorhynchus manningi_ (Okuno, 1996) | 6 |
| Infraorder Brachyura | | |
| Family Mithracidae | _Mithraculus forceps_ (A. Milne-Edwards, 1875) | 1 |
| **Order** Stomatopoda | | |
| Suborder Unipeltata | | |
| Family Gonodactylidae | _Neogonodactylus oerstedii_ (Hansen, 1895) | 3 |
| | _Neogonodactylus curacaoensis_ (Schmitt, 1924) | 1 |
| **PHYLUM CHORDATA** | | |
| **CLASS OSTEICHTHYES** | | |
| **Order Labriformes** | | |
| Family Labridae | _Thalassoma bifasciatum_ (Bloch, 1791) | 16 |
| | _Halichoeres bivittatus_ (Bloch, 1791) | 18 |
| | _Halichoeres garnoti_ (Valenciennes, 1839) | 1 |
| | _Halichoeres_ sp. (Rüppell, 1835) | 5 |
| Family Scaridae | _Nicholsina usta_ (Valenciennes, 1840) | 4 |
| | _Scarus taeniopterus_ (Desmarest,1831) | 3 |
| | _Sparisoma aurofrenatum_ (Valenciennes, 1840) | 1 |
| **Order Gobiiformes** | | |
| Family Grammatidae | _Gramma loreto_ (Poey, 1868) | 17 |
| Family Pomacentridae | _Chromis cyanea_ (Poey, 1860) | 1 |
| | _Stegastes partitus_ (Poey, 1868) | 4 |
| | _Stegastes_ sp. (Jenyns, 1840) | 6 |
| Family Opistognathidae | _Opistognathus_ sp. (Cuvier, 1816) | 2 |
| **Order Acanthuriformes** | | |

| Table 5 (continued) | | |
|---|---|---|
| **Taxa** | **Species** | **Fi** |
| Family Acanthuridae | *Acanthurus* sp. (Forsskål, 1775) | 3 |
| **Order Holocentriformes** | | |
| Family Holocentridae | *Neoniphon marianus* (Cuvier, 1829) | 2 |
| **Order Blenniiformes** | | |
| Family Chaenopsidae | *Lucayablennius zingaro* (Böhlke, 1957) | 1 |
| Family Labrisomidae | *Malacoctenus triangulatus* (Springer, 1959) | 1 |
| **Order Kurtiformes** | | |
| Family Apogonidae | *Zapogon evermanni* (Jordan & Snyder, 1904) | 1 |
| | *Apogon binotatus* (Poey, 1867) | 2 |
| **Order Perciformes** | | |
| Family Mullidae | *Pseudopeneus maculatus* (Bloch, 1793) | 1 |
| | *Mulloidichthys martinicus* (Cuvier, 1829) | 1 |
| Family Serranidae | *Serranus* sp. (Cuvier, 1816) | 7 |
| **Order Tetraodontiformes** | | |
| Family Monacanthidae | | 1* |
| **Order Syngnathiformes** | | |
| Family Aulostomidae | *Aulostomus maculatus* (Valenciennes, 1837) | 1 |

**Note:**
* In cases where the frequency of occurrence of a taxonomic category higher than species is recorded, it is because the entity could not be identified at a lower taxonomic level; Fi does not include species that have been identified and is also in that taxonomic category, it only refers to those identified at that taxonomic level.

## DISCUSSION

### Total length: spatial and temporal variation

Juvenile lionfish (total length smaller than 160 mm, FishBase (*Froese & Pauly, 2019*)) were caught in all the three regions (Fig. 2), suggesting that they are not restricted to a specific habitat at this stage. Although habitat type is likely exerting a great influence on the lionfish size, non-significant differences were mostly observed between habitats. For instance, R1 is an area with well-developed coral reefs with high substrate heterogeneity, a large spur and groove area and it is the region with the greatest ocean influence with a notable drop off. Then, conversely to our results, it was expected to find individuals with larger sizes in R1 since: (1) it has a greater number of refuges, (2) it possibly presents a greater abundance of prey and (3) relative abundant and larger sized lionfish usually inhabit deeper areas (*Pinheiro et al., 2016; Reed et al., 2018; Bustos-Montes et al., 2020*). It has been suggested that deep-water reef drop off habitats are beneficial for lionfish which has been discussed in other Cuban MPAs. In this regard, higher density and biomass of lionfish was found in the drop off habitat of Guanahacabibes National Park, probably due to the higher structural complexity, thus shelter available for lionfish (*Cobián-Rojas et al., 2016*). It has also been suggested that larger size in deeper-water residents could contribute to massive spawning and continued recruitment of lionfish on shallow reefs (*Bustos-Montes et al., 2020*).

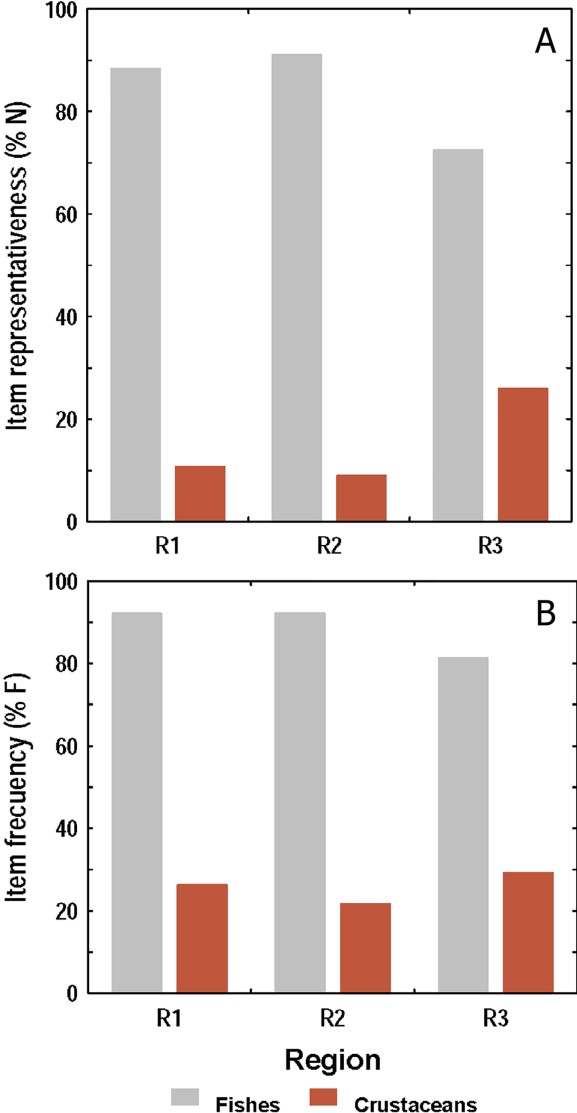

**Figure 5 Consumption of fishes and crustaceans by the lionfish analyzed according to the numerical method (A) and the frequency method (B) in three regions of the study area.**

However, some differences were observed in lionfish body size between sites, mainly those related to R3, which should be attributed to habitat characteristics. In a study conducted in the Turks and Caicos Islands, differences in lionfish sizes between deep coral reef (mean size: 227.0 mm) and seagrass bed (mean size: 150.0 mm) areas were attributed to ontogenetic habitat shift (*Claydon, Calosso & Traiger, 2012*). However, the results obtained in a subsequent investigation in the Bahamas, suggest that an ontogenetic habitat shift towards coral reefs does not occur in lionfish (*Pimiento et al., 2015*). Lionfish are usually scarce in the seagrass bed and mangroves, probably the mangrove roots are not a habitat they prefer; while in the seagrass bed they can be found in patch reefs shelters and in artificial shelters used in the lobster fishery (*Rodríguez-Viera et al., 2017*; and

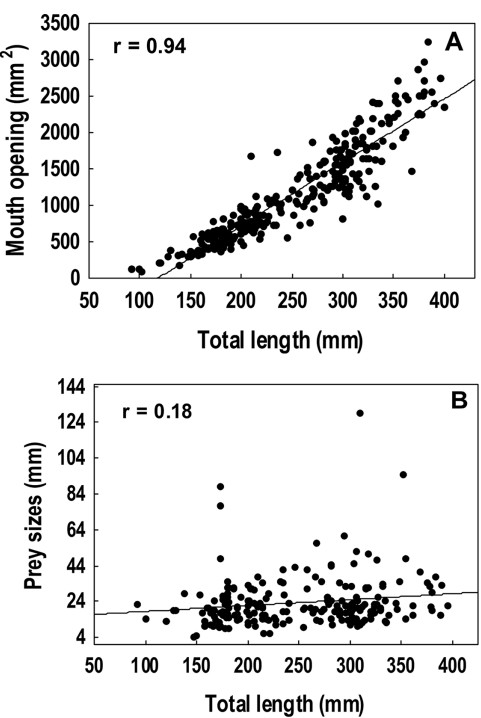

**Figure 6 Spearman rank correlation analysis among the mouth dimensions and the total length of the lionfish, r = 0.94 (A) and between the average size of the prey and the total length of the lionfish, r = 0.18 (B).**

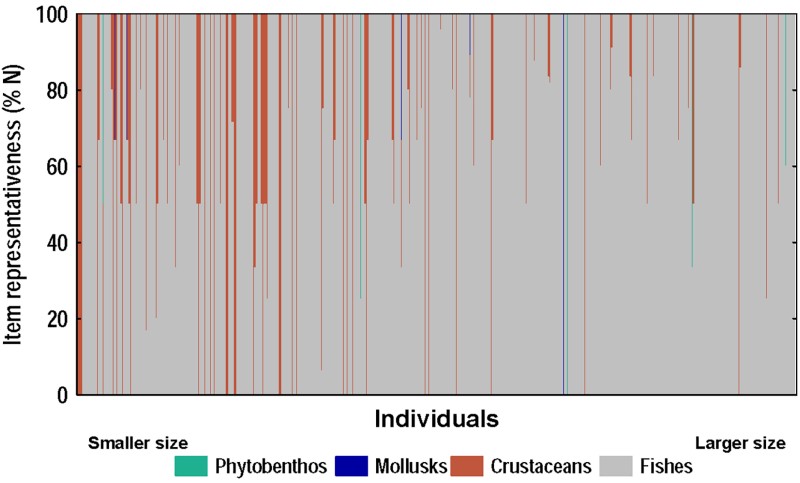

**Figure 7 Diet composition of lionfish analyzed according to the numerical method.** The x-axis shows the composition of the diet of each individual-ranked from smallest to largest according to their total length (total length range: 92–396 mm, $n = 367$).

observations made by the authors in the Cuban shelf). However, in the Cayos de San Felipe National Park smaller lionfish have been found in mangrove, suggesting they are using that habitat as nursery (*de la Guardia et al., 2017*), while in the Jardines de la Reina National Park this species has been reported abundant in coral reefs and mangrove areas (*Pina-Amargós, Salvat-Torres & López-Fernández, 2012*), both of which are Cuban MPAs.

Indeed, it would be very interesting to carry out a study that includes differences in depth and substrate complexity in all of these formations within and close to coral reefs, and particularly the contribution of typical "nursery" areas and deep reefs to the maintenance of lionfish populations.

Regarding the temporal variation of lionfish total length, the observed decrease trend is a positive result that could be evidencing the effectiveness of the lionfish extractions carried out in PFNP. The higher lionfish biomass caught during this study occurred in 2015 in R1 (226 individuals, biomass: 67.73 kg, mean total length: 259.52 mm), which could be one of the reasons that the following year it was detected a notable decrease in the average sizes of the specimens caught in that region (210.63 mm), and a general trend that continued over time. In addition, divers from the International Dive Center Marina Colony systematically have been extracting lionfish from Punta Frances coral reefs, since 2013 and mainly inside R1. The effects of lionfish catch on their abundance and sizes have been evidenced in various regions within the invaded Caribbean. For example, in the Bonaire Marine Park and the Florida Keys, systematic extractions have been made at dive sites since the beginning of the invasion, and a decrease in lionfish density and sizes has been observed compared to other nearby areas where this activity is not carried out (*Akins, 2013*). The removal of lionfish as a control strategy has also been implemented in Little Cayman, where a decrease in their total lengths was evident (*Frazer et al., 2012*). In other regions such as the Bahamas, Mexico and the Cayman Islands, control of lionfish at the local level has achieved a reduction in their densities that has remained stable over time (*Akins, 2013*). A study done in the Guanahacabibes National Park (Cuba) and the Arrecifes de Xcalak National Park (Mexico), showed a lower abundance of the lionfish in the second, attributed to systematic removals as part of a management program (*Cobián-Rojas et al., 2018*). These results suggest the efficiency of this strategy in reducing the direct impact of the invasive fish on the most vulnerable components of the ecosystem.

The tendency to reduce the lengths of lionfish over time is an encouraging result. This could indicate that despite reaching adulthood, lionfish are removed before reaching maximum lengths, which would mean that their longevity in PFNP is being reduced as well as their capacity to produce large number of eggs. The decrease in size could have a negative effect on their reproduction and contribute to reduce their abundance in the MPA. For instance, a study carried out in Little Cayman showed that fecundity is significantly lower in smaller female lionfish (*Gardner et al., 2015*). However, unfortunately, we lacked information for estimating lionfish population size structure in PFNP, and is not possible to provide additional evidence that support our discussion on size decrease and its causes over time. Further analyses using those additional data can provide new insights on this topic for this or other colonized regions.

## Length-weight relationship

By means of the length-weight relationship, it was possible to estimate the weight of the specimens that could not be weighed, and therefore to estimate the lionfish biomass extracted during the present investigation. This equation may have similar utility in other lionfish surveys. The usefulness of this type of equations has been evidenced in other

studies within the invaded Caribbean and western Atlantic (*Barbour et al., 2011*; *Darling et al., 2011*). Additionally, it allows estimating the condition factor of the species between different habitats and geographic regions (*Froese, 2006*), as has been used in lionfish (*Toledo-Hernández et al., 2014*; *Sabido-Itzá, Aguilar-Perera & Medina-Quej, 2016*). Moreover, it constitutes an important contribution in the Caribbean area, and particularly enriches the list of equations registered for fishes that inhabit Cuba (*Claro, Lindeman & Parenti, 2001*).

Our approach of one equation for two species is based on two aspects mainly. Firstly, *P. volitans* and *P. miles* are very similar morphologically (*Schultz, 1986*; *Morris et al., 2009*), being only appropriately differentiated in their non-native distribution range through molecular analysis (*Hamner, Freshwater & Whtifield, 2007*; *Morris & Whitfield, 2009*; *Burford-Reiskind et al., 2019*). Secondly, it is probable that the sampled individuals belong to the species *P. volitans*, which has been the species identified in the close Cuban MPA Guanahacabibes National Park (*Labastida et al., 2015*), and it has been more abundant than *P. miles* in the Greater Caribbean (*Hamner, Freshwater & Whtifield, 2007*; *Betancur-R et al., 2011*; *Guzmán-Méndez et al., 2017*). Therefore, we assume conservatively, that the equation provided in this study can be valid for *P. volitans*/*P. miles* populations in PNPF and closer areas. In addition, the relationship obtained in the present study, and therefore the constants *a* and *b*, are similar to those obtained for *Pterois* spp. in other invaded areas, *e.g.*, Little Cayman (*Edwards, Frazer & Jacoby, 2014*) and central Mexican Caribbean (*Villaseñor-Derbez & Fitzgerald, 2019*). It is similar to those found in other regions for *P. volitans* (*Froese & Pauly, 2019*) and for other species of the Scorpaenidae family (*Kulbicki et al., 1993*; *Froese & Pauly, 2019*).

## Lionfish diet characterization

Diet composition found on lionfish in this study was similar to what has been reported for other locations in Cuba (*Chevalier, 2017*; *Pantoja et al., 2017*) and in the Caribbean, *e.g.*, Bahamas, the southeastern United States, western Florida, Bonaire, and Mexico in terms of the zoological groups represented and the most abundant fish families (*Albins & Hixon, 2008*; *McCleery, 2011*; *Muñoz, Currin & Whitfield, 2011*; *Dahl & Patterson III, 2014*; *Santamaria, Locascio & Greenan, 2020*). Several of these studies also found predominance of fishes in the lionfish diet (*Albins & Hixon, 2008*; *Chevalier et al., 2008*; *Pantoja et al., 2017*; *Santamaria, Locascio & Greenan, 2020*), consumption of the same crustacean orders and high abundance of consumed Decapods order (*García, 2015*; *Pantoja, 2016*). Presence of phytobenthos in the lionfish diet has been explained by other researchers as accidental ingestion during the prey capture (*Pantoja et al., 2017*).

Fifteen fish families have been registered as the most abundant in the lionfish diet according to 11 studies done in the Caribbean region (five of them from Cuba) (Fig. 8). The present study identified 12 of these families, showing consistency with what has been reported for Cuba and the Caribbean. These findings suggest that lionfish tend to consume the same fish families in different locations in Cuba (*García, 2015*; *Pantoja, 2016*; *Chevalier, 2017*; *Pantoja et al., 2017*) and in other regions such as the Bahamas, the southeastern United States, Bonaire, Mexico, Belize, western Florida (*Morris & Akins,*

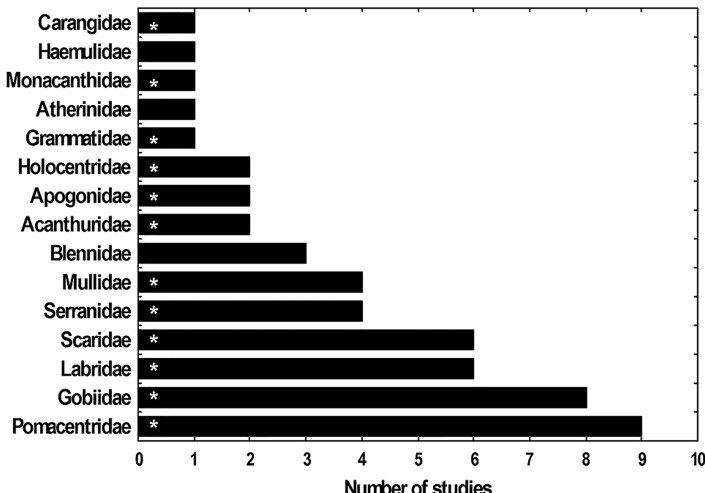

**Figure 8** **Frequency (Fi) with which several families of fishes appear as most abundant in studies of the diet of lionfish.** The families that were identified in this study are indicated with asterisk. The studies used to make the graph were: *Morris & Akins (2009)*, *Sandel (2011)*, *Muñoz, Currin & Whitfield (2011)*, *McCleery (2011)*, *Valdez-Moreno et al. (2012)*, *Cabrera (2014)*, *García (2015)*, *Pantoja (2016)*, *Hackerott et al. (2017)*, *Chevalier (2017)* and *Pantoja et al. (2017)*.

*2009*; *McCleery, 2011*; *Muñoz, Currin & Whitfield, 2011*; *Valdez-Moreno et al., 2012*; *Hackerott et al., 2017*; *Santamaria, Locascio & Greenan, 2020*). Such results could indicate a certain preference of the invasive fish for these preys and (or) their high availability. In this sense, the families that were most abundant in the present work agree with those reported by *Navarro-Martínez et al. (2021)* as the families with the greatest abundance and diversity in PFNP, therefore their high frequency of occurrence in the lionfish diet could be due to their high availability as prey in this area. A similar trend was observed with the three most abundant fish species in the stomachs analyzed (*H. bivittatus*, *G. loreto* and *T. bifasciatum*). Such results are consistent with the opportunistic feeding strategy of lionfish, which tend to feed on the most available prey (*Muñoz, Currin & Whitfield, 2011*; *Valdez-Moreno et al., 2012*; *Cobián-Rojas et al., 2016*).

Importantly, among the prey identified in the lionfish diet are those whose abundance could decrease due to predation by this exotic invasive species. This fact was evidenced by negative correlations between the abundance of lionfish and these species (*e.g.*, *T. bifasciatum, G. loreto, Stegastes partitus* and *Halichoeres garnoti*), recently obtained in the Guanahacabibes National Park, Cuba (*Cobián-Rojas et al., 2018*). In addition, herbivorous fishes (*Scarus taeniopterus, Sparisoma aurofrenatum* and *Acanthurus* sp.) are among those identified in the lionfish stomachs, which although not new, is an alarming result considering the importance of these key species for coral reefs. All these aspects demand future studies aimed at evaluating the possible effect of lionfish on coral reef ecosystem.

The R1 and R2 regions are found in reef areas close to each other, and have a similar species composition; therefore, it is not surprising that lionfish diet composition was not significantly different between the two regions. On the other hand, R3 comprises a

different habitat, *i.e.*, seagrass bed, which likely explains dietary differences of lionfish inhabiting this region compared to the coral reef-dominated R1 and R2. In seagrass areas, crustaceans tend to be very abundant and this would explain the results obtained in this and other works, where the consumption of shrimp tends to increase in the diet of lionfish caught in this type of ecosystem (*e.g. Chevalier et al., 2014*). Additionally, in coral reefs, crustaceans usually can reach larger sizes (*Nakamura & Sano, 2005*), so fewer may need to be eaten to meet lionfish energy needs. In contrast, small size crustaceans are abundant in seagrass beds, since this habitat constitutes a breeding area for this group of invertebrates (*Kenyon et al., 1999*), and probably the lionfish requires consuming a greater quantity of them. On the other hand, some crustaceans that inhabit coral reefs will feed on marine seagrass at night, which increases their availability to be consumed in these habitats. The lower crustacean consumption obtained by the numerical method with respect to those obtained by the frequency method, show that crustaceans are frequently present in the diet of lionfish, but they represent a minor fraction of the stomach content.

## Relationship between diet and morphometric features

The relationship between lionfish body size and gape size is approximately linear (Fig. 6). The increment of lionfish gape size with the individual total length allows lionfish to consume larger prey (*Rojas-Vélez, Tavera & Acero, 2019*). In the Bahamas, it was found that the average sizes of ingested fishes and crustaceans increased with lionfish size (*Morris & Akins, 2009*). However, in the present study, larger lionfish tended to consume prey of similar size to that of individuals of smaller sizes, despite having the capacity to consume larger prey. This behavior may be because smaller preys are more abundant in the study area (*Navarro-Martínez, 2015*; *Navarro-Martínez et al., 2021*), and therefore are more accessible to lionfish. For instance, during the coral reef ichthyofauna survey developed inside and outside PFNP between 2011–2014, most of identified fishes (63%) belonged to the families Pomacentridae and Labridae (*Navarro-Martínez, 2015*; *Navarro-Martínez et al., 2021*), whose sizes averaged 99.74 mm ($n$ = 2,241 measurements); while, total averaged size was 168.08 mm after the measure of 6,989 fishes (range: 36–726 mm), from which 1,530 were smaller than 100 mm (Z. Navarro-Martínez, 2011–2014, personal communications). However, we should be cautious with these data, since the methodology used by these authors (diver operated stereo-video) tend to underestimate very cryptic and small fishes, being the smaller (<35 mm) very difficult/unable to measure (*Navarro-Martínez et al., 2017*). Conversely, the ichthyofauna survey developed in the closer Siguanea inlet in 2015 and using seine net (closer to R3; *Rodríguez-Viera et al., 2017*), tended to more captures of smaller fishes which averaged ~55.26 mm ($n$ = 2,666 measurements, range: 10–400 mm) (L. Rodríguez-Viera, 2015, personal communications). Although the available data not provides conclusive evidence, since different methodologies were used in closed but different places to our survey sites, it is evident than small preys (smaller and similar to the preys detected in our samples of lionfish digestive content: ~45 mm) are abundant in all the area, but larger preys are also available.

In addition, we lacked of information about invertebrates assemblages from the study area, which also conform an important part of the lionfish diet, and they correspond to the

smaller preys. Concordantly, differences in local prey availability are suggested drivers of prey consumption patterns (*Ritger et al., 2020*). Lionfish appear to consume the most abundant prey regardless of taxonomic category and sizes; to the extreme of radically changing the composition of their diet and taking advantage of the most available preys, in the face of a lack of their most frequent preys (*Muñoz, Currin & Whitfield, 2011*; *Valdez-Moreno et al., 2012*; *Ritger et al., 2020*).

The trend of more frequent fish consumption as lionfish increases in size has been documented in other studies in Cuba (*García, 2015*; *Pantoja, 2016*), and in other invaded Caribbean regions (*Morris & Akins, 2009*; *McCleery, 2011*; *Muñoz, Currin & Whitfield, 2011*). Nevertheless, this trend is not always evident. *Chevalier (2017)* observed a trend between lionfish size and fish consumption, but not between lionfish size and crustaceans ingestion, while *Pantoja et al. (2017)*, did not detect a significant relationship between lionfish size and the intake of any of the two items.

In addition to prey availability in the study area, other factors are critical for lionfish abundance. For instance, lionfish potential predators, which can be its potential preys during juvenile stages, represent a critical issue to taking into account. In this regards, an ichthyofauna study in PFNP (*Navarro-Martínez et al., 2021*), showed that the genera that include the possible predators of the lionfish (*Epinephelus* and *Mycteroperca*), had low abundance, attributed to the inefficiency in PFNP protection mechanisms (*Angulo-Valdes & Hatcher, 2013*; *Navarro-Martínez, 2015*). Although groupers may be effective natural controllers of lionfish populations (*Mumby, Harborne & Brumbaugh, 2011*), their low abundance in PFNP rule out that possibility. On the other hand, some authors have found other factors non-related to groupers representativeness (*e.g.*, habitat characteristics and management strategies) as more related predictors of lionfish abundance (*Valdivia et al., 2014*). For instance, some metapopulation modeling approaches (*e.g.*, the incidence function model) have been useful to examine invasive populations, using a relatively simple model and field data (*e.g.*, size, dispersal, location, behavior) to generate information that helps managers the most effective management strategies (*Tamburello, Ma & Côté, 2019*). Future studies are required to continue the investigations related to the invasive lionfish in PFNP.

## CONCLUDING REMARKS

Aspects related to the sustained extraction of lionfish biomass during 4 years of sampling and its effect on the size of lionfish population in the wild were discussed. In this sense, a tendency to decrease of lionfish total length was observed over time, which could be due to the extractions carried out. This management and control strategy for lionfish in PFNP, although still insufficient, has had a positive effect on its control, and it must be maintained and improved. Therefore, it is suggested to maintain systematic extractions of lionfish as an adequate control strategy for MPAs with such characteristics, since decreasing the number of these predators and their average sizes, also reduces their predatory effect and the negative influence on the ecosystem.

The present work relates the composition of the lionfish diet with some of its morphometric variables and discusses general trends along different habitats of the

mangrove-seagrass bed-coral reef system that occur inside and adjacent to PFNP. Although larger lionfish are able to consume larger preys, we instead observed less dependence between prey and lionfish size consumption, since lionfish are likely feeding on the most available preys regardless of their sizes. In addition, there was a clear predominance of fishes in the diet of larger lionfish, while smaller individuals had higher proportions of crustaceans in their diet. The effect of lionfish predation over the more frequent consumed species and those with important ecological role, are a huge concern as additional stress to the currently affected marine ecosystem.

## ACKNOWLEDGEMENTS

The authors express their gratitude to the crew of the research vessel "Felipe Poey" for their assistance and the volunteers of the Operation Wallacea. Thanks to Armando Peréz and Lázaro V. García for their technical assistance. We thank the reviewers for their valuable comments on our MS.

### Funding

The authors received no funding for this work.

### Competing Interests

The authors declare that they have no competing interests.

### Author Contributions

- Laura del Río analyzed the data, prepared figures and/or tables, authored or reviewed drafts of the article, and approved the final draft.
- Zenaida María Navarro-Martínez analyzed the data, prepared figures and/or tables, authored or reviewed drafts of the article, and approved the final draft.
- Alexei Ruiz-Abierno conceived and designed the experiments, performed the experiments, analyzed the data, authored or reviewed drafts of the article, and approved the final draft.
- Pedro Pablo Chevalier-Monteagudo analyzed the data, authored or reviewed drafts of the article, and approved the final draft.
- Jorge A. Angulo-Valdes conceived and designed the experiments, performed the experiments, authored or reviewed drafts of the article, and approved the final draft.
- Leandro Rodriguez-Viera conceived and designed the experiments, performed the experiments, analyzed the data, prepared figures and/or tables, authored or reviewed drafts of the article, and approved the final draft.

### Animal Ethics

The following information was supplied relating to ethical approvals (*i.e.*, approving body and any reference numbers):

Committee of the Center for Marine Research at the University of Havana for animal care and use approved the study (Authorization code CIM/029).

## Field Study Permissions

The following information was supplied relating to field study approvals (*i.e.*, approving body and any reference numbers):

Lionfish were collected in Punta Frances MPA, Cuba, under all applicable local, state, and Cuban laws regulations; and the regulation of the Committee of the Center for Marine Research at the University of Havana for animal care and use.

## Data Availability

The raw measurements are available in the Supplemental Files.

## Supplemental Information

Supplemental information for this article can be found online at http://dx.doi.org/10.7717/peerj.14250#supplemental-information.

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
