# Peer review of "Feeding ecology of invasive lionfish in the Punta Frances MPA, Cuba: insight into morphological features, diet and management"

_PeerJ, doi:10.7717/peerj.14250_

## Round 0.1 · original submission · Major Revisions

I agree with both reviewers that although the manuscript is, in general, well structured, it needs additional revision based on the comments and suggestions of the reviewers. I look forward to reading the revised version of this interesting manuscript.

·

Basic reporting

Thanks a lot for the chance to review this interesting work. The manuscript is well structured, the introduction appropriately sets the scene for the following sections and the findings are presented and discussed in a compelling manner. I particularly enjoyed the flow of ideas that guide the reader through the text. I thus congratulate the authors for a good manuscript presenting interesting data on the feeding patterns of an invasive species in a Cuban MPA. Nonetheless I have some comments that should be addressed before the article can be accepted.

Specific comments:

1. Please explain why you did not use some kind of volume-based method to evaluate stomach content of lionfish in this study (one large prey item could correspond to several smaller prey items and equally meet the predator’s needs). I was thus expecting to see some kind of quantification beyond the counting of prey items. While you are stating that you measured the length of each prey item encountered (L143-145), you do not present any related data in the manuscript (and the raw data). In this context I was wondering, how you handled fragmented prey items (e.g. disintegrated shrimps) in this study?

2. Throughout the results section, you are reporting mean values for lionfish length. Please also report some measure of variability (e.g. standard deviation) alongside the means throughout the manuscript.

3. L77-80: While I agree with the general nature of this statement, your reasoning for why the study of morphometric variables is important is not convincing to me.

4. While you state that the maximum sampling depth was at 30m, it would be interesting to see the sampling depths for the captured lionfish (I couldn’t find any information in the raw data), or at least a bathymetric map of the study area that also indicates your sampling points. Please consider adding something like this in the supplementary files.

5. Your survey took place from 2013 to 2016 (June, July and August of each respective year). The supplied field permit, however, indicates a time period from January to April 2016 and thus does not cover the period in which this study took place.

6. The last author is also the one who gives ethical approval for this study. Please explain why this is the case.

7. You introduce the acronym PFNP. However, I do not understand what the acronym “PFNPA” in L123 stands for.

8. L166-167: Not clear to me what you mean with this statement. Please clarify / rephrase.

9. What is the reasoning to set the size range for the comparison of diet composition at 100 - 352mm? The upper limit does not correspond to the largest size of individuals you report and the lower limit is lower than the reported size limit for juvenile lionfish. The setting of this size range thus seems arbitrary to me. Please explain and elaborate in the manuscript. Potentially also specify how many sampled individuals were removed from the analysis based on this criterion.

10. L183: I believe you are referring to Spearman rank correlations (cf. “ranges”)?

11. L204: Why are you not reporting R2 in 2015 here?

12. L214: Why did you include “168g” in this line? Remove or clarify?

13. L215: I don’t understand why you are reporting 307 individuals (in 2015) here, while you report 310 individuals for 2015 in the previous sentence. What happened to these three individuals?

14. Fi is sometimes referred to as frequency of appearance and sometimes as frequency of occurrence. Please consider standardizing.

15. Please use kilograms (instead of grams) when reporting total biomass or aggregated biomass per area (e.g. in Tab. 4 and throughout the manuscript).

16. L232: Please specify what you mean with “remaining families”.

17. L258: Please reference the respective figure here.

18. Throughout the manuscript, you are sometimes using “y” to connect the names of authors of publications with two authors (e.g. “Albins y Hixon 2008). Please update this according to journal criteria.

19. You are hinting upon the fact that the removal of ~600 individuals over a period of 4 years significantly reduced mean lengths of lionfish in the study area. Do you have any idea / data on which fraction of the population of lionfish in the whole MPA or wider area this constitutes (e.g. estimate of population density, estimate of total number of lionfish in the area, …?) that could help you to substantiate this argument?

20. Concerning the length-weight relationship (Fig. 4), you are reporting one equation for two species. Please elaborate in the manuscript why this is a valid approach and reference accordingly.

21. L414-415: Please specify which results you are referring to in this statement.

22. L432-434: Please rephrase for clarity.

23. L438-439: Which data support this argument? Do you have any data / information on the availability and size structure of prey items (see comment below)?

24. Do you have access to data on accompanying fauna (e.g. densities and length-frequency distributions of prey species), which could support your argument of opportunistic feeding and shed light on why larger lionfish don’t eat larger prey (L395-396, e.g. Navarro-Martinez, 2015?). I am wondering if, in the study area, larger prey items would generally be available to larger lionfish or if prey items don’t grow larger than you observed in the stomachs? If larger prey items are in fact available in sufficient densities, then I invite you to speculate on the reason why larger lionfish do not consume them.

25. L439-440: Please rephrase.

26. Please check all references to Figures and Tables in the manuscript. For example, in L225, you are referring to Table 4, while presenting data found in Table 5.

27. From Table 5, I deduce that you only found and identified a total of ~130 prey items in the stomachs of lionfish. However, according to the raw data you identified 1247 prey items in these stomachs. Please appropriately explain the reason for this difference in reporting.

As a general comment, I suggest to re-work the text for English language clarity in order to improve comprehension by an international audience. Some examples where writing could be improved include lines L22, L29, L43, L46, L67, L71, L95, L111, L119, L136, L175 (and more). Furthermore, there are a few grammatical errors throughout the text which can be easily flattened out. I suggest you have a fluent English speaker read over your manuscript to improve its clarity and readability.

I compliment you for gathering and compiling an interesting dataset for the Punta Frances MPA spanning several years. I thank you for providing the raw data upon which the manuscript is based and appreciate your honesty regarding the lost sample identifiers. In summary, I very much enjoyed reading your manuscript, which tells a coherent and interesting story. There are, however, some weaknesses (see specific comments above), which should be improved before acceptance.

All the best for the revision and I’m looking forward to seeing an improved version of your manuscript!

Experimental design

See specific comments under point 1.

Validity of the findings

See specific comments under point 1.

Reviewer 2 ·

Basic reporting

Clear usage of scientific and professional English throughout the manuscript. Sufficient background and references are provided for Introduction and Discussion. Research questions are well defined and within the scope of the journal.

Experimental design

1. Please include descriptions of R1, R2 and R3 (whether it is seagrass, coral reef and/or mangrove areas) in Figure 1 and figure caption.

2. Line 122-123: "The R3 region is seagrass bed habitat located outside PFNP, and it was only sampled in 2015." The authors need to provide further justification for including R3 in the dataset since it is located outside PFNP and only sampled once. Furthermore, the difference in significance seems to influence results and discussion.

Validity of the findings

3. It will be interesting if the authors could highlight or group the dataset by sexual maturity as mentioned in Line 198-199. It may be a reason for the decrease in the size of lionfish between years and/or regions. It also highlights the efficiency of systematic removal by the PFNP management.

Additional comments

Discussion:
4. Line 217-294: These two paragraphs discussed the same topic which is the "ontogenetic habitat shift" and thus can be shortened and combined.

Conclusion:
5. Line 421-434 discussed possible predators for the lionfish. In my opinion, it should be in Discussion section under subsection "Relationship between diet and morphometric features ".

Annotated reviews are not available for download in order to protect the identity of reviewers who chose to remain anonymous.

---

## Round 0.2 · accepted · Accept

Thank you for addressing all the comments from the reviewers. I agree with the reviewer that this manuscript is ready for publication.

Reviewer 2 ·

Basic reporting

As per my last review.

Experimental design

As per my last review.

Validity of the findings

As per my last review.

Additional comments

The authors have addressed all my concerns and I am satisfied with their replies. I would thus recommend this manuscript be accepted for publication.